# Pediatric post-discharge mortality in resource-poor countries: A protocol for an updated systematic review and meta-analysis

Maryum Chaudhry[1,2], Martina Knappett[2], Vuong Nguyen[2], Jessica Trawin[2], Nathan Kenya Mugisha[3], Jerome Kabakyenga[4], Elias Kumbakumba[5], Shevin Jacob[3,6], J. Mark Ansermino[2,7], Niranjan Kissoon[2,8], Matthew O. Wiens[2,3,7,9]*

1 Department of Health Sciences, Simon Fraser University, Burnaby, British Columbia, Canada, 2 Centre for International Child Health, British Columbia Children's Hospital, Vancouver, British Columbia, Canada, 3 Walimu, Kampala, Uganda, 4 Department of Community Health, Mbarara University of Science and Technology, Mbarara, Uganda, 5 Department of Pediatrics, Mbarara University of Science and Technology, Mbarara, Uganda, 6 Department of Clinical Sciences, Liverpool School of Tropical Medicine, Liverpool, United Kingdom, 7 Department of Anesthesiology, Pharmacology and Therapeutics, University of British Columbia, Vancouver, British Columbia, Canada, 8 Department of Pediatrics, University of British Columbia, Vancouver, British Columbia, Canada, 9 Mbarara University of Science and Technology, Mbarara, Uganda

* matthew.wiens@bcchr.ca

**Data Availability Statement:** No datasets were generated or analysed during the current study. All

## Abstract

### Background

More than 50 countries, mainly in Sub-Saharan Africa and South Asia, are not on course to meet the neonatal and under-five mortality target set by the Sustainable Development Goals (SDGs) for the year 2030. One important, yet neglected, aspect of child mortality rates is deaths occurring during the post-discharge period. For children living in resource-poor countries, the rate of post-discharge mortality within the first several months after discharge is often as high as the rates observed during the initial admission period. This has generally been observed within the context of acute illness and has been closely linked to underlying conditions such as malnutrition, HIV, and anemia. These post-discharge mortality rates tend to be underreported and present a major oversight in the efforts to reduce overall child mortality. This review will explore recurrent illness following discharge through determination of rates of, and risk factors for, pediatric post-discharge mortality in resource-poor settings.

### Methods

Eligible studies will be retrieved using MEDLINE, EMBASE, and CINAHL databases. Only studies with a post-discharge observation period of more than 7 days following discharge will be eligible for inclusion. Secondary outcomes will include post-discharge mortality relative to in-hospital mortality, overall readmission rates, pooled estimates of risk factors (e.g. admission details vs discharge factors, clinical vs social factors), pooled post-discharge mortality Kaplan-Meier survival curves, and outcomes by disease subgroups (e.g. malnutrition, anemia, general admissions). A narrative description of the included studies will be

relevant data from this study will be made available upon study completion.

**Funding:** The author(s) received no specific funding for this work.

**Competing interests:** The authors have declared that no competing interests exist.

synthesized to categorize commonly affected patient population categories and a random-effects meta-analysis will be conducted to quantify overall post-discharge mortality rates at the 6-month time point.

## Discussion

Post-discharge mortality contributes to global child mortality rates with a greater burden of deaths occurring in resource-poor settings. Literature concentrated on child mortality published over the last decade has expanded to focus on the fatal outcomes of children post-discharge and associated risk factors. The results from this systematic review will inform current policy and interventions on the epidemiological burden of post-discharge mortality and morbidity following acute illness among children living in resource-poor settings.

## Systematic review registration

PROSPERO Registration ID: CRD42022350975.

## Introduction

Though significant gains in child survival have been achieved in recent decades, the global burden of mortality remains primarily concentrated in resource-poor countries. Most deaths in these regions are attributable to causes that are largely preventable and generally treatable, such as acute respiratory infections, diarrheal illness, and malaria [1]. It is increasingly recognized that many childhood deaths occur during the post-discharge period. Indeed, many studies suggest that mortality rates following discharge are similar to those seen during the hospitalization period [2, 3].

Despite growing evidence emphasizing the importance of pediatric post-discharge mortality research, practice and policy-level efforts to address current challenges in discharge and post-care have not seen a commensurate degree of interest. The causal factors related to a weak response to this crisis in pediatric care are complex but one important factor is that current research has remained largely fragmented, in terms of underlying population subgroups (e.g. malnutrition, anemia, general admissions), outcome details (e.g. post-discharge time frame, post-discharge relative to in-hospital mortality) and risk factors (e.g. admission details vs discharge factors, clinical vs social factors). Although prior systematic reviews exist, the paucity of prior research has precluded robust meta-analytic pooling within these key disease subgroups [4, 5]. Since the publication of these prior reviews, a significant number of key studies have been published. These studies provide the critical data needed for more sophisticated modelling of pooled post-discharge mortality and readmissions data. A clear and robust summary of the burden of post-discharge mortality is urgently required in order to inform a concerted public health response.

The aim of this study is to update the current evidence base on the epidemiological burden and related risk factors of post-discharge mortality following acute illness among children living in resource-poor settings. We will conduct an updated systematic literature review to identify all studies published since the previous systematic review performed in 2018 [5]. With these additional studies we will conduct a meta-analysis to provide a more generalizable estimate of the burden of post-discharge mortality, as well as the specific risk factors for mortality, within the relevant population subgroups.

## Materials and methods

### Objectives

To determine the rates of, and risk factors for, pediatric post-discharge mortality following discharge from hospitals in resource-poor settings.

### Protocol and registration

This protocol was developed in accordance with the Preferred Reporting Items for Systematic Reviews and Meta-Analyses protocols (PRISMA-P) checklist (S1 File) [6]. The protocol for this systematic review was registered on the PROSPERO database with registration number: CRD42022350975.

### Eligibility criteria

The eligibility criteria used will be in accordance with the CoCoPop framework which is recommended for use in reviews of etiology studies [7]. CoCoPop represents condition, context, and population.

**Population.** The population of interest is pediatric patients 0–18 years of age discharged from hospitals in developing countries (as defined under "Context") following acute illness. Studies with no pediatric data will be excluded, including those where pediatric data cannot be distinguished from adult data. Patient populations discharged from settings other than publicly accessible hospitals (e.g. military hospitals) will be excluded. Studies representing a specific non-infectious disease population where post-discharge care and outcomes would likely be different than that following acute (primarily infectious) illness including specific congenital diseases (e.g. cardiac), cancer, surgical populations, or other specific non-infectious admissions including trauma, kidney disease, cardiac disease, ophthalmic disease, sickle cell disease, liver disease, epilepsy, burns, poisoning, asthma, or prematurity will be excluded. Conditions that often exist as comorbidities within this population will be examined as risk factors.

**Condition.** Mortality occurring following hospitalization due to an acute illness.

**Context.** The context for eligible studies to be identified through the systematic literature search will encompass the publication window of January 1, 2017 to January 31, 2023. All studies published prior to 2017 will be identified from an earlier systematic review with similar search strategies and will supplement the final pool of included studies. Only studies conducted in developing countries will be included; developing countries will be defined as those countries classified by the United Nations Development Programme as having a low or low-middle Socio-Demographic Index (SDI) as of 2019, as well as those countries included in the previous reviews as having a low Human Development Index in 2011 and 2016 (S4 Table) [8–10].

Eligible study designs include randomized control trials (RCTs), prospective or retrospective cohort studies, and studies using surveillance/registry data. Studies with no post-discharge outcomes reported beyond 7 days, or if the discharge was following a non-admission (e.g. following birth) will be excluded. Finally, unpublished studies, studies where the full text is unavailable, or studies not providing original data will be excluded. The minimum sample size for inclusion in the meta-analysis is n = 100 patients.

### Outcomes

The primary outcome will be the pooled prevalence of pediatric post-discharge mortality in resource-poor settings six months post-discharge. Secondary outcomes will include post-

discharge mortality relative to in-hospital mortality, pooled estimates of risk factors, overall readmission rates, and pooled post-discharge mortality Kaplan-Meier survival curves.

## Data selection and search strategy

MEDLINE, EMBASE, and CINAHL databases will be searched between the dates of January 1, 2017 to January 31, 2023. Studies published prior to 2017 were identified in a previous review and will supplement the final pool of studies [5]. In collaboration with two academic librarians, two investigators (MK and MC) developed search strategies for each of the three databases of interest (S1–S3 Tables). Briefly, subject headings related to post-discharge mortality, such as "hospitalization" or "mortality" are exploded, in addition to keywords such as "hospital*" or "death*". The search strategy further includes subject headings and keywords related to longitudinal or follow-up studies to improve the specificity of the results. Low and low-middle SDI country names and general developing country terms were included as keywords [11–13]. The population of interest is detailed for MEDLINE using a validated pediatric search filter with subject headings including "adolescent" or "child" and keywords including "infant*" or "child*" [14]. This filter is adjusted for the EMBASE and CINAHL databases and uses similarly relevant keywords. Our search strategy excludes inapplicable publication types, animal studies, and studies concentrating on neoplasms. There is no language restriction in place.

## Study selection and screening process

Two investigators (MK and MC) will screen titles and abstracts according to the study inclusion criteria using Covidence systematic review management software [15]. Selected abstracts will then undergo full-text review by both reviewers for inclusion into the study. Any conflicts will be resolved by the Principal Investigator (MW). Finally, the references of the final pool of studies will be searched for any additional studies that meet the inclusion criteria. A PRISMA flow diagram will be used to illustrate the screening process [16]. Searches will be re-run prior to the final analysis.

## Data extraction and quality assessment

Data will be extracted by two investigators (MK and MC) in Microsoft Excel. A full extraction template is included in the supplementary data (S5–S8 Tables). Extraction will include both study characteristics (design, duration, sample size, location, etc.) and outcome characteristics (outcome type, outcome rate, outcome risk factors, etc.) up to one-year post-discharge. For randomized control trials, both the control arm and intervention arm(s) will be combined if the study finds no significant difference in post-discharge mortality between the two; if a significant difference is present, only data from the control arm will be retrieved. Extraction of risk factors associated with post-discharge mortality upon admission and discharge, as well as for re-admission will comprise of multivariate and/or univariate analyses. Kaplan-Meier survival curves, where provided, will be extracted using a plot digitizer [17]. Quality assessment will be completed using the a customized critical appraisal tool which has been adapted from the JBI critical appraisal tools (S9 Table). Each study will be independently assessed by two investigators. Any disagreements will be resolved through discussion. If a consensus cannot be reached, the PI will make the final decision. Publication bias will be assessed by visual inspection of asymmetry in the funnel plots, Egger's test for asymmetry and the trim-and-fill method to correct for funnel plot asymmetry [18]. This will be considered in the context of likely study heterogeneity which may affect interpretation of the funnel plot [19].

## Quality assessment

Quality assessment will be completed using a 7-item checklist, based on the JBI critical appraisal tool, which has been adapted to suit the specific requirements of this review (S9 Table) [20]. Each study will be independent assessed by two investigators and discrepancies will be resolved through discussion. If a consensus cannot be reached, the PI will make the final decision.

## Assumptions

In cases where multiple publications draw on the same sample, data from each study will be extracted as a single study. In cases of studies partially drawing on the same study sample, if there is more than 50% of a population overlap, and where non-overlapping data is not separately reported, data will only be extracted for the larger cohort. This assumes that the larger cohort will also include the children analyzed in the smaller cohort. The denominator used for assessing post-discharge mortality will include all children enrolled for follow-up at discharge, including any children lost to follow-up prior to 6 months post-discharge. The children lost to follow-up in this case are, therefore, assumed to be alive, which may underestimate post-discharge mortality rates. Studies which enrolled all acute illness admissions will be used to estimate overall post-discharge mortality prevalence. If a study cohort only consisted of individuals with a particular disease (e.g., severe anemia), they will be included in the meta-analysis of disease subgroups but not used in the estimate of overall post-discharge mortality.

## Data synthesis and analysis

A narrative description of the included studies will be organized into patient populations represented in the data (e.g., respiratory infections, malaria, malnutrition). Data will be quantitatively synthesized if a sufficient number of unique studies are selected to enable a meta-analysis.

Both fixed and random-effects meta-analysis will be conducted for estimating overall post-discharge mortality at 6 and 12 months. When pooling prevalence, children who were lost to follow-up will be included in the denominator. Studies may report risk factors for post-discharge mortality as either hazards ratios, relative risks, or odds ratios depending on the study design and objectives. These different outcome measures will be pooled and analyzed separately. Additionally, these measures may have been calculated using univariate analysis (unadjusted) or multivariate analysis adjusting for other potential risk factors or covariates. Estimates from unadjusted and adjusted analyses will be pooled and analyzed using a mixed-effects meta-regression model with the inclusion of a covariate indicating whether the estimate was from an unadjusted or adjusted analysis. Forest plots will be used to illustrate mortality rates at specific time points and/or hazard ratios for risk predictors. The variance of the hazards ratios for risk factors will be estimated from the 95% confidence intervals [21]. Heterogeneity will be measured using the $I^2$ statistic, where an $I^2$ value of 0% to 25% will be categorized as low heterogeneity, 50% to 75% will be moderate heterogeneity, and 75% to 100% will indicate substantial heterogeneity [22].

A distribution-free approach assuming random effects will be used to pool survival curves to estimate the pooled risk of post-discharge mortality over time up until 12 months [23]. In addition, survival regression assuming a Gompertz distribution will be fit to each survival curve to extrapolate survival up until 12 months followed by the Combescure method for pooling survival curves. The numbers at risk at each time point for pooling survival curves will be estimated according to the methods described in Tierney et al. using code made publicly available by Guyot et al. with the minimum amount of information needed being the initial sample

size at time-point 0 [21, 24]. The unit of time was converted to weeks if the curves were reported in a different time unit and survival curves were presented with the percentage of mortality on the y-axis.

Additional analysis will include examining pooled outcomes based on disease subgroup (e.g., all admissions, malnutrition, respiratory infection, diarrheal diseases, malaria, anemia) or world regions, as defined by the World Health Organization [25]. For outcomes not synthesized, they will be reported descriptively for each individual study or across all studies.

All analysis of data will be conducted using R Statistical Software with the *metafor* package for meta-analysis, the *metaSurvival* package for pooling survival curves, and the *flexsurv* package for survival regression [26–28].

## Anticipated results

The results from this systematic review will add to, and summarize, the current evidence base on the epidemiological burden of post-discharge mortality following acute illness among children living in resource-poor settings. Though we expect the results to remain consistent with prior reviews, this meta-analysis will add significantly to prior data synthesis projects by reporting the pooled 6 and 12-month post-discharge mortality estimates for all key population subgroups, pooled risk factors for mortality within these subgroups, as well as a formal assessment of the distribution in-hospital vs post-discharge mortality as proportions of overall mortality. For all population subgroups, we will report both study-level and pooled estimates for each outcome. We will also report pooled Kaplan-Meier survival curves, which provides important information about the timing of mortality during the post-discharge period.

## Discussion

Pediatric post-discharge mortality has been previously shown to be a key contributor to overall child mortality in resource-poor settings, though until recently, few studies have been conducted to explore this important problem. Current policy and improvement in practice to address post-discharge mortality lag behind a rapidly growing body of evidence highlighting the severity of the problem and core population subgroups with unique post-discharge vulnerabilities. This systematic review and meta-analysis will provide pivotal summary evidence of this issue as it currently exists in resource-poor nations. The review will highlight the severity and risk factors for post-discharge mortality within key population subgroups which is crucial for developing evidence-based and resource-appropriate post-discharge interventions. This work will ultimately be used to justify improvements to existing child mortality healthcare strategies in resource-poor settings that include pediatric post-discharge mortality.

This review is subject to several limitations. It specifically focuses on resource-poor settings, for which there is no ideal definition. To maximize consistency in interpretation, the research team chose SDI as an indicator of a country's development status; SDI calculates life expectancy, education level, per capita income, and the impact of ecological factors on a nation that influence the progression of human development [29]. We believe that this indicator best captures the country-level data representing settings where post-discharge mortality remains a key issue. Nevertheless, over time countries can move between SDI levels, and thus our search only captures countries currently listed as low or low-middle status. Moreover, as resource limitations are not uniformly distributed within any one country, SDI may not reflect the true spectrum of development as it does not account for this heterogeneity.

In addition, this review is designed as an update of a prior systematic review [5]. The most recent search strategy only encompassed the previous five years, and thus may not include studies which were not identified during the prior systematic review. However, formative

studies that might have been missed through the search are likely to be captured through reference searching of included studies and review of past and present reviews. Moreover, given that the same research team is conducting this review, and the expert knowledge within our team in this subject area, we do not believe any crucial studies were missed during the prior review.

Heterogeneity in study results is anticipated and can be attributed to multiple factors, including diversity present in sample populations, post-discharge follow-up duration, the observational nature of the research, and geographic diversity. To partly address this, similar study populations will be pooled, and a random effects modelling approach will be used. Additionally, heterogeneity statistics will be reported for individual interpretation. There may also be differences in how categories of severity are defined for various risk factors, such as age, malnutrition and anemia. This may further add to estimate heterogeneity and may also limit the ability to pool certain risk factors.

Studies may also merge pediatric cases of mortality with non-pediatric populations, failing to stratify results by age. As a result of merging data, the total number of studies selected for extraction may be fewer in number. Moreover, longitudinal studies which report all-cause mortality without distinguishing in-hospital and post-discharge deaths will not be eligible. This is a common method of reporting in many randomized controlled trials and will, therefore, limit the data available. Lastly, by primarily using databases where the bulk of studies are written in English, studies published in non-English databases might be omitted. Despite these limitations, this literature review will strengthen the existing evidence to contribute a more generalizable estimate of the burden of post-discharge mortality in resource-poor settings.

## Conclusion

As post-discharge mortality is an important contributor to child mortality, this systematic review and meta-analysis will provide new data as to its burden in resource-poor settings. Through this data synthesis project, clinicians, researchers and policy makers will be further equipped to work towards the development and implementation of solutions to improve the hospital-to-home transition, especially for those at highest risk of poor outcomes.

## Supporting information

**S1 File. PRISMA-P checklist.**
(DOC)

**S1 Table. Search strategy Ovid MEDLINE.**
(DOCX)

**S2 Table. Search strategy Ovid EMBASE.**
(DOCX)

**S3 Table. Search strategy EBSCO CINAHL.**
(DOCX)

**S4 Table. List of countries classified as low and low-middle SDI.**
(DOCX)

**S5 Table. Extraction template for study characteristics.**
(DOCX)

**S6 Table. Extraction template for outcome characteristics.**
(DOCX)

**S7 Table. Extraction template for post-discharge mortality timeline.**
(DOCX)

**S8 Table. Extraction template for post-discharge mortality risk factors.**
(DOCX)

**S9 Table. Quality appraisal checklist.**
(DOCX)

## Author Contributions

**Conceptualization:** Matthew O. Wiens.

**Investigation:** Maryum Chaudhry, Martina Knappett.

**Methodology:** Maryum Chaudhry, Martina Knappett, Matthew O. Wiens.

**Project administration:** Maryum Chaudhry, Martina Knappett.

**Supervision:** Matthew O. Wiens.

**Writing – original draft:** Maryum Chaudhry, Vuong Nguyen, Matthew O. Wiens.

**Writing – review & editing:** Maryum Chaudhry, Martina Knappett, Vuong Nguyen, Jessica Trawin, Nathan Kenya Mugisha, Jerome Kabakyenga, Elias Kumbakumba, Shevin Jacob, J. Mark Ansermino, Niranjan Kissoon, Matthew O. Wiens.

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
