## [Decision Letter · Decision Letter 0]

9 Jan 2023

PONE-D-22-28459

Pediatric post-discharge mortality in resource-poor countries: A protocol for an updated systematic review and meta-analysis

PLOS ONE

Dear Dr. Wiens,

Thank you for submitting your manuscript to PLOS ONE. After careful consideration, we feel that it has merit but does not fully meet PLOS ONE’s publication criteria as it currently stands. Therefore, we invite you to submit a revised version of the manuscript that addresses the points raised by reviewer 2 during the review process.

We look forward to receiving your revised manuscript.

Kind regards,

Andrea L. Conroy, PhD

Academic Editor

PLOS ONE

Journal Requirements:

Reviewers' comments:

Reviewer's Responses to Questions

**Comments to the Author**

1. Does the manuscript provide a valid rationale for the proposed study, with clearly identified and justified research questions?

Reviewer #1: Yes

Reviewer #2: Yes

2. Is the protocol technically sound and planned in a manner that will lead to a meaningful outcome and allow testing the stated hypotheses?

Reviewer #1: Yes

Reviewer #2: Yes

3. Is the methodology feasible and described in sufficient detail to allow the work to be replicable?

Reviewer #1: Yes

Reviewer #2: Yes

4. Have the authors described where all data underlying the findings will be made available when the study is complete?

Reviewer #1: Yes

Reviewer #2: Yes

5. Is the manuscript presented in an intelligible fashion and written in standard English?

Reviewer #1: Yes

Reviewer #2: Yes

6. Review Comments to the Author

You may also provide optional suggestions and comments to authors that they might find helpful in planning their study.

Reviewer #1: This is a review of "Pediatric post-discharge mortality in resource-poor countries: A protocol for an updated systematic review and meta-analysis" by Chaudhry M et al. Overall I found the protocol paper to be well written and I have no major comments to offer.

Reviewer #2: PONE-D-22-28459

Article Type: Study Protocol

Full Title: Pediatric post-discharge mortality in resource-poor countries: A protocol for an updated systematic review and meta-analysis.

This is a straightforward article describing the protocol for a systematic review and meta-analysis of post-discharge pediatric mortality in resource-poor countries. As such, I believe it has the merits to be published, and have only a few comments to suggest, in order to make it clearer.

• Abstract: “largely due to preventable causes such as infectious diseases”. I would add “…or chronic conditions such as malnutrition”.

• Line 58-59: may be useful adding a timeframe “Indeed, many studies suggest that mortality rates following discharge are similar to those seen during the hospitalization period”. Perhaps add “…in the few weeks post-discharge”

• What is the rationale of excluding deaths in the first 7 days post discharge? These are also relevant, and although very closely linked to the acute event, they may still require different approaches for prevention. Would include them (perhaps as a secondary outcome?) or would include those that occur after a deliberate decision of discharge (rather than after abscondment).

• One of the key elements on the problem of post discharge mortality is how those deaths are ascertained, so as not to incur on specific biases. For this purpose, sites that do have DSS (demographic surveillance system) and that can ensure an adequate follow up of the vast majority of discharged patients are better than those who don’t have it. Perhaps this consideration (method by which patients were followed up for outcome ascertainment) should be included as a piece of data to collect

• Given the effort that researchers are going to make, would it be worth collecting prevalence of overt neurological sequelae by month 6 also as an additional outcome?

7. PLOS authors have the option to publish the peer review history of their article (what does this mean?). If published, this will include your full peer review and any attached files.

Reviewer #1: No

Reviewer #2: **Yes: **Quique Bassat

---

## [Author Response · Author response to Decision Letter 0]

27 Jan 2023

Reviewer 1 

Comment 1: Overall I found the protocol paper to be well-written and I have no major comments to offer.

Response: Thank you. We appreciate the time you took to review the manuscript. 

Reviewer 2

Comment 1: Abstract: “largely due to preventable causes such as infectious diseases”. I would add “…or chronic conditions such as malnutrition”.

Response 1: Thank you for this helpful suggestion. We agree that this sentence does not adequately capture the context of the existing literature. The manuscript has been revised accordingly. 

The abstract now states: “For children living in resource-poor countries, the rate of post-discharge mortality within the first several months after discharge is often as high as the rates observed during the initial admission period. This has generally been observed within the context of acute illness and has been closely linked to underlying conditions such as malnutrition, HIV, and anemia.”

Comment 2: Line 58-59: may be useful adding a timeframe “Indeed, many studies suggest that mortality rates following discharge are similar to those seen during the hospitalization period”. Perhaps add “…in the few weeks post-discharge”

Response: We agree that adding additional time-context is appropriate. The adjusted sentences are pasted in the response to comment 1. 

Comment 3: What is the rationale of excluding deaths in the first 7 days post discharge? These are also relevant, and although very closely linked to the acute event, they may still require different approaches for prevention. Would include them (perhaps as a secondary outcome?) or would include those that occur after a deliberate decision of discharge (rather than after abscondment). 

Response: Thank you for highlighting this. We agree that early deaths are of critical importance, and indeed we do intend to include them in our analysis. However, as a means of capturing studies relevant to this systematic review, we will only include studies which include an observational period of more than 7 days. This is to ensure a sufficient proportion of studies which capture data on our primary outcome, which is 6-month mortality. We agree that the way we have framed this is confusing and have adjusted our wording accordingly. 

The amended sentence is as follows: “Only studies with a post-discharge observation period of more than 7 days following discharge will be eligible for inclusion.”

Comment 4: One of the key elements on the problem of post discharge mortality is how those deaths are ascertained, so as not to incur on specific biases. For this purpose, sites that do have DSS (demographic surveillance system) and that can ensure an adequate follow up of the vast majority of discharged patients are better than those who don’t have it. Perhaps this consideration (method by which patients were followed up for outcome ascertainment) should be included as a piece of data to collect.

Response: You have raised an important consideration here and we agree with this comment. The method and proportion of follow-up are critical considerations for study quality. Although it is not explicitly stated in the main text of the manuscript, the extraction template in “S5 Table. Extraction Template for Study Characteristics” includes a column for collecting data on the post-discharge follow-up method.

Comment 5: Given the effort that researchers are going to make, would it be worth collecting the prevalence of overt neurological sequelae by month 6 also as an additional outcome?

Response: Thank you for this suggestion. We agree that neurological outcomes are important. To our knowledge this is very rarely reported in the post-discharge mortality literature. We have amended our data extraction template to include this outcome, but have not included this in the text of the manuscript as we anticipate too few studies to complete any formal analysis on this outcome. The prevalence of the reporting of this outcome, however, will be an interesting result which perhaps can spur more work in this area. 

Alterations to Reference List

Reference #1: The previous citation was of a website that summarized the Levels and Trends in Child Mortality 2021 Report released by UN IGME. UN IGME has since released a Levels and Trends in Child Mortality 2022 Report resulting in concurrent changes on the website. To avoid difficulties in locating information resulting from future website updates, the most recent report was located and replaced the previously cited website. 

Reference #8: The Global Burden of Disease webpage previously linked opened to the incorrect dataset. To remedy this, the correct website link was inserted. 

Reference #8: A secondary document was originally linked in the citation. The original Human Development Report 2011 report was located and replaced initially cited summary report. 

Reference #9: Similar to reference #8, a secondary document was linked. The original Human Development Report 2016 was located and replaced the initially cited statistical annex.

---

## [Editor Report · Decision Letter 1]

31 Jan 2023

Pediatric post-discharge mortality in resource-poor countries: A protocol for an updated systematic review and meta-analysis

PONE-D-22-28459R1

Dear Dr. Wiens,

We’re pleased to inform you that your manuscript has been judged scientifically suitable for publication and will be formally accepted for publication once it meets all outstanding technical requirements.

Kind regards,

Andrea L. Conroy, PhD

Academic Editor

PLOS ONE
---

## [Editor Report · Acceptance letter]

14 Feb 2023

PONE-D-22-28459R1 

Pediatric post-discharge mortality in resource-poor countries: A protocol for an updated systematic review and meta-analysis 

Dear Dr. Wiens:

I'm pleased to inform you that your manuscript has been deemed suitable for publication in PLOS ONE. Congratulations! Your manuscript is now with our production department. 

Kind regards, 

on behalf of

Dr. Andrea L. Conroy 

Academic Editor

PLOS ONE